# An Observational Study of Skeletal Malformations in Four Semi-Intensively Reared Carp Species

**DOI:** 10.3390/vetsci11010030

**Published:** 2024-01-12

**Authors:** Caterina Varvara, Edmond Hala, Mariasevera Di Comite, Rosa Zupa, Letizia Passantino, Gianluca Ventriglia, Angelo Quaranta, Aldo Corriero, Chrysovalentinos Pousis

**Affiliations:** 1Department of Veterinary Medicine, University of Bari Aldo Moro, 70010 Valenzano, Italy; caterina.varvara@uniba.it (C.V.); rosa.zupa@uniba.it (R.Z.); angelo.quaranta@uniba.it (A.Q.); aldo.corriero@uniba.it (A.C.); c.pousis@ibiom.cnr.it (C.P.); 2Department of Aquaculture and Fisheries, Faculty of Agriculture and Environment, Agricultural University of Tirana, 1030 Tirana, Albania; hiedmo@ubt.edu.al; 3Department of Basic Medical Sciences, Neurosciences and Sensory Organs, University of Bari Aldo Moro, 70124 Bari, Italy; mariasevera.dicomite@uniba.it; 4Department of Precision and Regenerative Medicine and Jonian Area (DiMePRe-J), Section of Veterinary Science and Animal Production, University of Bari Aldo Moro, 70010 Valenzano, Italy; letizia.passantino@uniba.it; 5Institute of Biomembranes, Bioenergetics and Molecular Biotechnologies, National Research Council, 70126 Bari, Italy

**Keywords:** skeletal anomalies, carp development, carp rearing

## Abstract

**Simple Summary:**

Carps include some of the most widely farmed fresh water species. In the present study, the incidence of skeletal malformations in the fingerlings of four carp species reared in semi-intensive conditions in Albania was assessed. The investigated species were the common carp (*Cyprinus carpio*), silver carp (*Hypophthalmichthys molitrix*), grass carp (*Ctenopharyngodon idella*) and bighead carp (*Hypophthalmichthys nobilis*). The incidence of skeletal malformations and body growth were much higher in the herbivorous silver carp, grass carp and bighead carp than in the omnivorous common carp. We hypothesized that (i) the feeding protocol, which included ingredients of animal origin, may have caused an abnormal increase in the body growth rate associated with an increase in skeletal malformations in the herbivorous carp species; (ii) more efficient broodstock management may help increase egg quality and reduce skeletal malformations.

**Abstract:**

Cyprinids include some of the most widely farmed freshwater species. The aim of this study was to assess the incidence of skeletal malformations in carp species reared in semi-intensive systems in Albania: common carp (*Cyprinus carpio*), silver carp (*Hypophthalmichthys molitrix*), grass carp (*Ctenopharyngodon idella*) and bighead carp (*Hypophthalmichthys nobilis*). The common carp fingerlings had a mean total length of 28.9 ± 5.0 mm; the frequencies of individuals with at least one anomaly and at least one severe anomaly were 79.2% and 43.4%, respectively. The silver carp juveniles had a mean total length of 21.6 ± 2.1 mm; the frequencies of individuals with at least one anomaly and one severe anomaly were 93.1% and 57.5%, respectively. The grass carp fry had a mean total length of 33.5 ± 2.6 mm; all the analyzed specimens showed almost one anomaly and 86.4% showed at least one severe anomaly. The bighead carp juveniles had a mean total length of 34.4 ± 5.7 mm; the frequencies of individuals with at least one anomaly and at least one severe anomaly were 95.0% and 62.5%, respectively. The development of a more suitable feeding protocol for herbivorous species and the setting up of more efficient broodstock management protocols are suggested to reduce the high incidence of skeletal malformations.

## 1. Introduction

Skeletal anomalies are frequently reported in cultured fish and affect fish welfare, growth and health [1,2,3,4]. Skeletal abnormalities are frequently associated with slow body growth and various types of co-morbidities, resulting in a high mortality rate that increases production costs, compromises product marketability and reduces farm competitiveness [4,5]. In particular, individuals affected by severe skeletal anomalies that modify the external morphology must be discarded, causing a variable loss of profit for the aquaculture industry [6]. The economic impact of skeletal malformation on fish farming industry is difficult to quantify due to the reluctance of producers to collect and release data in order to protect the farm’s reputation and avoid the loss of customers’ confidence [6]. 

To mitigate the commercialization of individuals affected by malformations, in the Mediterranean hatcheries, personnel manually remove malformed fry in the early stages (usually early juveniles with a mean weight around 0.3–0.5 g), and a second removal is made of juveniles with mean weights between 1 and 5 g [6]. However, manual selection based on the visual inspection of the fingerling shape is not always an effective method to reduce the incidence of malformed individuals, as minor undetected malformations may worsen as growth proceeds and eventually affect the entire production chain [6]. In toto staining protocols have been set up for a more efficient diagnosis of early juvenile skeletal malformations which are based on the differential staining of cartilage and bone by means of Alcian Blu and Alizarin Red, respectively [6,7]. This method is widely used in scientific studies on fish skeletal malformations but has limited applications in aquaculture because the staining procedure requires a few days of laboratory work. Recently, a rapid method to detect skeletal malformation through a microradiographic procedure has been proposed [8]. 

Cyprinids include some of the most widely farmed freshwater species that are recognized as an important source of animal protein for humankind [9]. Global carp aquaculture production of about 33,000 tons was recorded in 2018, with Asian countries being the main producers [10]. In Europe, the most widely farmed carp species is the common carp (*Cyprinus carpio*), which represents 6% of the total aquaculture production and is mainly produced by Eastern countries [10]. Other cyprinid species of aquacultural interest are the silver carp (*Hypophthalmichthys molitrix*), the grass carp (*Ctenopharyngodon idella*) and the bighead carp (*Hypophthalmichthys nobilis*) [11]. 

The total aquaculture production in Albania in 2021 was 8641 tonnes, to which marine aquaculture contributed with 6780 tonnes and inland aquaculture with 1861 tonnes [12]. Nowadays, carp are the second most produced aquaculture products after rainbow trout (*Oncorhynchus mykiss*). Carp production is aimed at supporting the economy of local rural areas, and it is carried out by public and private entities according to rearing protocols adapted according to the availability of raw ingredients for feed production [13]. The four existing carp hatcheries have an annual fingerling production not sufficient to cover the assessed needs of about 8.4–9.2 million carp fingerlings per year required to restock inland waters [14]. 

This observational study provided an interpretation of the different rate of skeletal malformations in the four carp species, namely common carp, bighead carp, grass carp and silver carp. The carps were produced at the carp hatchery of Tapiza (Fushë Krujë, Albania), and released in ponds for growing under semi-intensive conditions using a rearing protocol standard for the four species.

## 2. Material and Methods

### 2.1. Sampling and Double in Toto Staining

For the present observational study, 53 common carp, 86 silver carp, 44 grass carp and 40 bighead carp fingerlings, produced during routine activity in the didactic experimental hatchery of Tapiza, which is part of Agricultural University of Tirana, were sampled on 17 July 2022. The common carp fingerlings were derived from the natural spawning of hatchery-produced specimens, whereas all the other fingerlings of the other species were derived from eggs obtained from induced spawning of broodstocks treated with common carp pituitary extracts. Spawning occurred in May 2022 for common carp and June 2022 for all other species. Fertilized eggs were incubated in a prepared pond for the common carp, and in large jars incubators of 110 L at a density of about 900 eggs L^−1^ [15]. Soon after hatching, larvae were supplied with hardboiled chicken eggs; then, three to five days after hatching, the fish were transferred to artificial ponds fertilized with chicken organic manure (each species in a dedicated pond) at a density of about 500 larvae m^−3^. Pond oxygen saturation and salinity were 99% and 0.18 ppt, respectively. In ponds, fish diets were initially represented by the available phytoplankton and zooplankton; after a week, for the following three weeks, additional food consisting of a mixture of equal parts of soybean meal, wheat meal, fish meal and meat meal, in the form of a very fine dry powder made of 0.1–0.2 mm particles, was added. At the end of this period, at the age of about one month, fingerlings of bighead carp, grass carp and silver carp were sampled, while fingerlings of common carp were transferred to a larger pond where their diet was further supplemented with cereal powder containing 30% protein (50% of animal origin and 50% of vegetal origin) until they were sampled at two months of age. It was not possible to obtain food samples to determine proximate composition. 

Carp fingerlings were collected by hand net and euthanized with a lethal dose of anesthesia (0.4 mL/L of 2-phenoxyethanol, Merck, Rahway, NJ, USA). The samples were then fixed in 10% buffered formalin for 48 h at 4 °C, washed in phosphate-buffered saline (pH 7.5) for 48 h at 4 °C and stored in 70% ethanol. In the laboratory, samples were hydrated in 50% ethanol for 24 h and processed for double staining according to the procedure previously reported by [16] and modified by [8,17]. The samples were (i) immersed in a solution of 0.5% KOH and 3% H_2_O_2_ and exposed to sunlight for 4–6 h; (ii) incubated for 90 min in the dark in a solution containing 60 mL 95% ethanol, 40 mL acetic acid and 25 mg alcian blue; (iii) washed in 0.5% KOH; (iv) stained in 0.03% Alizarin Red solution (30 mg Alizarin Red in 100 mL of 0.5% KOH) for 1 h in the dark; (v) washed in 10 mL of 0.5% KOH and (vi) bleached by consecutive immersions in 0.5% KOH:glycerol 3:1; 0.5% KOH:glycerol 1:1; 0.5% KOH:glycerol 1:3; 100% glycerol). The first two steps lasted about 48 h; the third step was repeated until the dyes ceased to be released from the sample. 

### 2.2. Stereoscopic Analysis 

The samples were analyzed under a stereomicroscope (Leica WILD M3C, Wetzlar, Germany) equipped with a camera (DFC 420; Leica Microsystems, Cambridge, UK). An identification code was assigned to each specimen before undergoing microscopic analysis. The samples were analyzed three times by the same operator to limit potential mistakes in the identification and classification of abnormalities. The anatomical terminology of the present study complies with that previously reported [18,19,20,21]. Skeletal anomalies affecting vertebral centra and those affecting the external shape of the fish body (i.e., lordosis and kyphosis) were classified as severe, whereas the other anomalies were classified as mild [18].

For each fish, the total length (TL, in mm) was recorded, and the number of each type of anomaly observed was registered into binary matrices (Appendix A). The following variables were evaluated: relative frequency (%) of individuals with at least one anomaly; relative frequency (%) of individuals with at least one severe anomaly; ratio (%) of observed severe anomalies on the total number of observed anomalies; number of types of anomaly observed; total anomaly load (number of total anomalies/number of malformed individuals); severe anomaly load (number of severe anomalies/number of individuals with severe anomalies); relative frequency (%) of each type of anomaly, with respect to the total number of anomalies; relative frequencies (%) of individuals affected by each type of anomaly, in each group. 

### 2.3. Statistics

Statistical differences between the common carp and the other three examined species were assessed through the chi-squared test for all parameters, except differences in total anomalies load and severe anomaly load that were assessed through one-way ANOVA.

## 3. Results 

The biometric data and general data on the anomalies identified in all the examined specimens are reported in Table 1. 

In the common carp fingerlings, the mean TL of which was 28.9 ± 5.0 mm, 151 skeletal anomalies belonging to 23 different types were observed. In the silver carp juveniles, the mean TL of which was 21.6 ± 2.1 mm, 1029 skeletal anomalies belonging to 26 different types were recorded. In the grass carp juveniles, the mean TL of which was 33.5 ± 2.6 mm, 1148 skeletal anomalies belonging to 24 different types were found. In the bighead carp fingerlings, the TL of which was 34.4 ± 5.7 mm, 414 skeletal anomalies belonging to 20 different types were registered.

The frequency (%) of each type of anomaly, with respect to the total number of anomalies, is reported in Table 2; the frequency (%) of individuals affected by each type of anomaly is reported in Table 3, and the regional distribution of the different anomalies is reported in Table 4. 

In general, the common carp fingerlings showed lower rates of anomalies than the silver carp, grass carp and bighead carp. The total frequency of common carp individuals affected by at least one anomaly and at least one severe anomaly was 78.4% and 43.4%, respectively (Table 1). Compared with the common carp, all the other examined species had a significantly higher frequency of individuals affected by at least one anomaly (silver carp, 93.1; grass carp, 97.7%; bighead carp, 95%; *p* < 0.05 in all comparisons); grass carp and bighead carp had a significantly higher frequency individuals affected by at least one severe anomaly (grass carp, 86.4%; bighead carp, 62.5%; *p* < 0.05 in both comparisons). The total and the severe anomaly load of the common carp fingerlings were 3.6 and 2.9, respectively (Table 1). Compared with the common carp, all the other examined species had a significantly higher total anomaly load (silver carp, 12.7; grass carp, 26.1; bighead carp, 10.9; *p* < 0.01 in all comparisons); the silver carp and grass carp had a significantly higher severe anomaly load than the common carp (silver carp, 5.7; grass carp, 7.4; *p* < 0.05 in both comparisons). 

In the common carp, most of the anomalies were recorded in the caudal region (48.2% of total anomalies), followed by the haemal region (24.5%). In the other species, the most deformed area was the pre-haemal vertebral region (silver carp, 43.8%; grass carp, 40.4%; bighead carp, 39.4%), followed by the cephalic vertebral region in the silver carp (26.4%) and grass carp (25.3%) and by the caudal vertebral region in the bighead carp (24.2%) (Table 4). In the common carp, the frequency of deformed vertebral bodies in the caudal and haemal vertebral region and the frequency of anomalies of the neural and haemal arches (lack of fusion of the two laminae or fusion of laminae not belonging to the same vertebra) in the caudal vertebral region were higher compared with the other species (*p* < 0.05 in all comparisons). In contrast, the frequencies of anomalies affecting the neural arches in all the other body regions were lower than in the other species (*p* < 0.05 in all comparisons). 

The majority of the examined silver carp, grass carp and bighead carp fingerlings had at least one abnormal neural arch; consequently, the frequency of individuals affected by this anomaly in all the examined body areas was statistically higher (*p* < 0.01) compared with the common carp individuals.

The frequencies of common carp individuals with deformed vertebral centra in the cephalic and pre-haemal regions were statistically lower than in the silver and grass carp individuals (*p* < 0.05 in both comparisons). The frequencies of common carp individuals affected by deformed vertebral centra in the haemal and caudal regions were significantly higher than in the silver carp (*p* < 0.05).

The frequency of fused vertebral centra in the pre-haemal region was lower in the common carp fingerlings compared with all the other carp species (*p* < 0.05); moreover, the common carp fry had a lower frequency of fused vertebral centra in the haemal region compared with the grass carp juveniles (*p* < 0.05).

Representative micrographs of the recorded anomalies of the four analyzed carp species are reported in Figure 1. The Alizarin Red staining of vertebrae appeared more intense in the common carp fingerlings, suggesting higher mineralization, compared with the other analyzed species.

## 4. Discussion

In Mediterranean marine hatcheries, severe skeletal anomalies generally affect 30–40% of the fish production [7,22], resulting in significant economic loss for the fish farming industry [23]. Since malformed fish in the wild have a higher mortality than normally developed individuals [24], little information is available on the incidence of skeletal anomalies in wild fish populations. Comparative studies on gilthead sea bream (*Sparus aurata*) [1], sea bass (*Dicentrarchus labrax*) [25], sharpsnout sea bream (*Diplodus puntazzo*) and pandora *(Pagellus erythrinus*) [26,27], and Senegalese sole (*Solea senegalensis*) [24], indicate a higher incidence of severe abnormalities and variations in the number of meristic characters in reared fish than in wild populations. The data on skeletal anomalies in wild populations are even more limited for freshwater fish species, and to our knowledge, no information is available of the occurrence of skeletal anomalies in wild carp fingerlings. 

It has been reported that a 100 to 500 times higher incidence of malformations occurs in hatchery-produced fish than in wild populations of ayu sweetfish (*Plecoglossus altivelis*) [28]. In hatchery-produced zebrafish (*Danio rerio*), significant differences in the meristic characters but not in the incidence of skeletal anomalies has been reported [29]. 

The occurrence of skeletal deformities is believed to be related to environmental [23], genetic [7] and nutritional [2,4,30] factors. 

The present study was conducted on fish produced in a semi-intensive aquaculture system operated as an artisanal activity, the aim of which is to support the economy of rural areas. The production protocol used in this aquaculture activity may change year by year according to budget and raw material availability. The four carp species analyzed in the present study were reared according to the same rearing protocol that did not take into account species-specific nutritional requirements. In fact, only the common carp showed a growth rate similar to that reported in other studies, whereas the other species showed higher growth rates compared with those reported in the available literature. In the present study, the common carp at two months of age had a mean TL of 21.5 ± 3.7 mm, which was in the size ranges reported for individuals reared in aquaria (20.5–32.4 mm at 59 days) [31], and slightly lower than the size reported by [32] (28.3–30.4 mm at 56 days). At one month of age, the grass carp fingerlings of the present study had a mean TL of 33.5 ± 2.6 mm, a much larger size than those reported by [33] (17.5–20.6 mm at 56 days) and by [34] (19.2–25.3 mm at 70 days) for individuals reared in tanks under a natural photoperiod. The mean size of the bighead carp sampled in the present study at one month of age was 34.4 ± 5.7 mm TL, whereas [35] reported an overall mean body length of 15.56 mm in bighead carp fingerlings reared in aquaria and subjected to different feeding regimes and sampled 74 days after fertilization. An exceptionally high growth rate, similar to that recorded in the present study, was reported in bighead carp sampled at 28 days after fertilization and after 21 days of feeding *Artemia* nauplii or freshwater rotifer diets in recirculating and static systems [36]. In these complex experiments, fish were fed three to six times per day and reached body lengths between 31.2 and 41.7 mm when reared at low density (12.5 fish L^−1^), and 26 to 37.6 mm when reared at high density (50 fish L^−1^). 

A high growth rate is not always a good indicator of the final product quality, and a supra-physiological growth rate may be associated with increased skeletal anomalies [37] and decreased product quality [7]. The growth pattern of bone tissues varies according to the season and is affected by environmental conditions (temperature) and feeding. A fast body growth may affect the ossification process, resulting in a higher incidence of skeletal anomalies [38] and an abnormal acceleration of body growth may also result in reduced bone mineralization. Commonly used methods for age determination in fishes inhabiting temperate waters are based on the presence of bone discontinuities, which are interpreted as seasonal events [39,40,41]. These discontinuities are due to the alternation of opaque areas (corresponding to rapid summer growth) and translucent areas (corresponding to slower autumn-winter growth). The optical differences between translucent and opaque growth marks have been assumed to be related to different calcium concentrations, higher in the translucent ones, which are formed during the winter slow growth [40,42]. Finally, an inappropriate swimming speed during intensive growth could substantially increase red muscle development, and therefore results in strong mechanical forces (pressure) leading to vertebral tissue transformation [43,44,45]. In the present study, based on Alizarin Red affinity, grass carp, bighead carp and, to a lesser extent, silver carp, showed lower vertebral mineralization compared with common carp and the lower mineralization was associated with a higher growth rate and higher incidence of skeletal anomalies. The frequency of common carp fingerlings affected by at least one severe anomaly (43.2%) was similar to the reported range of marine farmed fish, which is generally around 30–40% [6,22]. The frequency of individuals with at least one severe anomaly observed in silver carp (57.5%), grass carp (86.4%) and bighead carp (62.5%) was much higher. The most frequent skeletal anomalies in the present study were those affecting the vertebral column (93.5–99.4%) and involved vertebral fusion and malformation, whereas a very low incidence of curvature anomalies was observed in all examined species. Moreover, the most affected vertebral region in the common carp was the caudal one, whereas most of the anomalies affected the pre-haemal region in the other examined species. Except for the numerous laboratory experiments carried out to assess the effects of experimental diets (e.g., [46,47]) or toxic compounds (e.g., [48,49]) on zebrafish skeletogenesis, limited information on the occurrence of skeletal anomalies in juvenile cyprinids is available. In a comparative study on juvenile zebrafish, a higher percentage (93.4%) of reared individuals have been reported to be affected by skeletal anomalies compared with wild-caught fish (87.2%) [29]. Among the skeletal anomalies reported in this zebrafish study, axis deviations and cephalic anomalies were rare, whereas vertebral body anomalies, including vertebral fusion, and various kinds of anomalies of vertebral elements (neural and haemal arches) were the most represented anomalies. Al-Harbi (2001) [50] reported that 24.9% of eight-month-old common carp reared in tanks were affected by different types of skeletal malformations. In ten-week-old crucian carp (*Carassius carassius* L.) reared in aquaria and fed natural feed (frozen Chironomidae sp. larvae), Kasprzak et al. (2017) [51] reported no skeletal anomalies. However, 40–60% of the individuals fed commercially available basic starter diets were affected by scoliosis and/or kyphosis.

According to [52], rearing in intensive conditions may be responsible for deformities of the skeletal elements of the caudal region, including vertebra centra, neural and haemal processes in sea bream fry, whereas skeletal anomalies of the cephalic/pre-haemal regions as well as of the anterior haemal region of the vertebral column were reported in juvenile Atlantic halibut (*Hippoglossus hippoglossus*) fed oxidized fish oil [30]. As usual in extensive carp farming systems [15], the fish of the present study were supplied with chicken eggs during the first days after hatching, then they fed from the natural pond production and were supplemented with artificial food. The use of chicken eggs, thanks to their content in phospholipid classes and particularly in phosphatidylinositol, has been proven to reduce skeletal anomalies in common carp larvae [53]. To our knowledge, no information of the effect of chicken eggs on strictly herbivorous carp species is available; however, grass carp have a limited capacity to utilize high dietary lipids, and an excessive intake of highly unsaturated fatty acids has been reported to result in several kinds of adverse effects [54]. In the present study, all the four carp species were reared according to the same protocol; therefore, a possible explanation for the observed different incidences of skeletal malformations is that the same feeding protocol used for fish having different nutritional requirements resulted in different rates of body growth and skeletal anomalies. In the wild, the fish species investigated in this study have different diets: the common carp is an omnivore species that feeds on a variety of plant and animal species, including amphipods, phantom midge larvae, mollusks and ostracods [55]; the grass carp is an herbivorous species feeding mainly on macrophytes [56]; the silver carp and the bighead carp are planktivorous species that feed on lower trophic level organisms [57,58]. 

In the rearing ponds, the sampled fish fed initially only on phytoplankton and zooplankton and, from the second week, they fed on a mixture of equal parts of soybean meal, wheat meal, fish meal and meat meal. During the second month, the common carp (the only species sampled at two months of age) diet was further supplemented with cereal powder containing 30% protein. Therefore, it may be hypothesized that the feeding protocol was more suitable for the omnivorous common carp, but it may have been inappropriate for the planktivorous carp species since it contained animal ingredients that may have induced an abnormally high growth rate associated with a high incidence of skeletal anomalies. The amino acid feed content has been reported to have an effect on the quality of fish larvae, since severe effects on larval development are associated with dietary imbalances in amino acids [59,60]. In particular, vertebral anomalies were observed in rainbow trout fed a diet containing high concentrations of leucine [61]. Since animal proteins contain more leucine than vegetal proteins [62], the administration of chicken eggs soon after hatching, and the addition of fish and meat ingredients to the cereal powder offered after the first week post-hatching, might have been responsible for the high incidence of vertebral anomalies observed in the herbivorous carp species.

Inadequate amounts of vitamins and macrominerals have been reported as possible causes of skeletal malformations in fishes [7]. However, an insufficient feed content of vitamins or macrominerals is generally associated with a reduced growth rate [7], whereas the growth rate of the herbivorous species of the present study was higher than expected. 

It cannot be excluded that the results of the present study may be related to sub-optimal broodstock management. In the present study, no information on inbreeding, a known cause of skeletal malformations [6], is available. Moreover, the common carp fingerlings originated by eggs obtained by spontaneous spawning, whereas the reproduction of the other three carp species was obtained using pituitary extract. Eggs obtained through the hormonal induction of spawning have sometimes been found to be smaller and provided with a smaller lipid droplet compared with eggs derived from spontaneous spawning [63], and lower fertilization, hatching and survival rates have been reported after the hormonal induction of spawning [63,64]. Since skeletogenesis alterations may originate during the embryonic phase [37], it cannot be excluded that the high incidence of skeletal malformations reported in the present study were caused by a low quality of the eggs spawned after the administration of pituitary extracts. Since high egg incubation temperatures have been reported to induce skeletal deformities in several fish species [38,65,66,67], another possible cause of the higher incidence of skeletal malformation may be related to the ambient temperature. The common carp fingerlings analyzed in the present study originated from eggs produced in May (maximum daily temperature ranging between 18 °C and 35 °C [68]), whereas the analyzed fingerlings of the other carp species originated from eggs spawned in June (maximum daily temperature ranging between 25 °C and 36 °C [68]). 

Regardless of the causative factors of the observed high incidence of skeletal anomalies in herbivorous carp, we hypothesize that the high incidence of skeletal anomalies in herbivorous carp species results in a higher mortality rate and product loss. In fact, in the present study, the skeletal anomalies mainly involved the vertebral centra (deformed or fused centra) and vertebral arches. According to [69], the spinal condition is the most important predictor of the swimming ability, and seriously deformed neural and haemal arches can affect blood flows and spinal cord functions.

In conclusion, the present study suggests that the feeding protocol used for the different carp species analyzed in the present study was more appropriate for the omnivorous common carp than for the herbivorous carp species, since it might have been responsible for the observed higher incidence of skeletal anomalies. Moreover, an improvement in broodstock management and a modification of the spawning induction treatment aimed at obtaining fertilized eggs when ambient temperatures are lower is proposed. 

## Figures and Tables

**Figure 1 vetsci-11-00030-f001:**
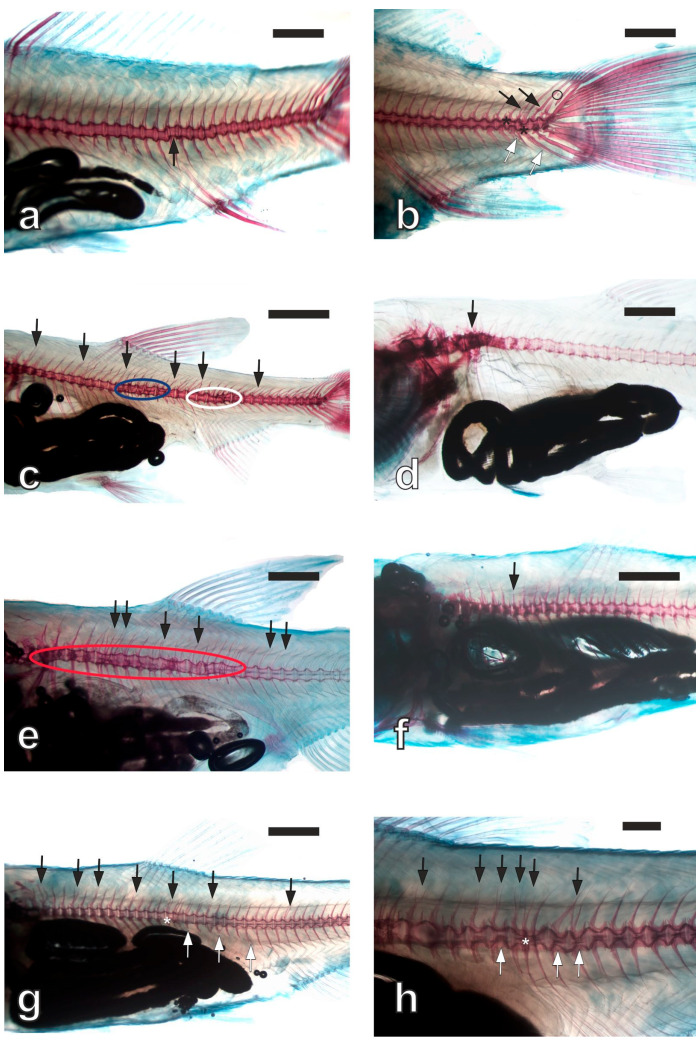
Representative skeletal anomalies observed in carp fingerlings. (**a**) Common carp; black arrow: deformed bodies of 21st to 24th vertebrae. (**b**) Common carp; black arrows: anomalous neural spines in the caudal region; black asterisks: deformed vertebral bodies in the caudal region; black circle: supernumerary bone in the caudal region; white arrows: anomalous haemal spines in the caudal region. (**c**) Silver carp; black arrows: multiple anomalous neural spines along the entire spinal column; blue oval outline: vertebral fusions and deformed vertebral bodies in the pre-haemal region; white oval outline: vertebral fusions and deformed vertebral bodies in the haemal region. (**d**) Silver carp; black arrow: kyphosis in the cephalic vertebrae region. (**e**) Grass carp; black arrows: multiple anomalous neural spines along the entire vertebral column; red oval outline: several vertebral fusions and deformed vertebral bodies in cephalic and pre-haemal regions. (**f**) Grass carp; black arrow: slight lordosis in the pre-haemal region. (**g**) Bighead carp; black arrows: multiple anomalous neural spines along the entire vertebral column; white arrows: anomalous haemal spines in the pre-haemal region; white asterisk: vertebral fusion in the pre-haemal region. (**h**) Bighead carp; black arrows: multiple anomalous neural spines in the pre-haemal and haemal regions; white arrows: deformed vertebral bodies in the pre-haemal and haemal regions; white asterisk: vertebral fusion in the pre-haemal region. In toto double staining method. Magnification bars: 1 mm in (**a**,**b**,**d**,**e**,**g**); 2 mm in (**c**,**f**); 0.5 mm in (**h**).

**Table 1 vetsci-11-00030-t001:** Number of fish examined, body length and general data on the identified anomalies.

	Common Carp	Silver Carp	Grass Carp	Bighead Carp
**Number of individuals**	53	86	44	40
**Total length (mm)**	28.9 ± 5.0	21.4 ± 2.1	32.8 ± 2.6	33.7 ± 5.7
**Number of anomalies recorded**	151	1119	1148	414
**Relative frequency (%) of individuals with at least one anomaly**	78.4	93.1	100.0	95.0
**Total anomalies load**	3.6	12.7	26.1	10.9
**Relative frequency of individuals with at least one severe anomaly (%)**	43.4	57.5	86.4	62.5
**Ratio (%) of observed severe anomalies on the total number of observed anomalies**	43.7	27.9	24.5	22.0
**Severe anomalies load**	23.0	5.7	7.4	3.6

**Table 2 vetsci-11-00030-t002:** Relative frequencies (%) of each type of anomaly.

Region	Type of Anomaly	Common Carp	Silver Carp	Grass Carp	Bighead Carp
**Cephalic vertebrae**	Kyphosis	0.0	0.3	0.3	0.0
	Lordosis	2.0	0.6	0.3 **	0.0 **
	Vertebral fusion	0.7	3.6	2.1	0.2
	Vertebral anomaly	0.7	3.3	2.3	0.5
	Anomalous neural arch and/or spine	6.6	18.1 **	20.3 **	10.4
	Anomalous haemal arch and/or spine	0.7	0.6	0.1	0.0
**Pre-haemal vertebrae**	Scoliosis	0.0	0.1	0.0	0.0
	Kyphosis	1.3	0.3	0.0 **	0.0 *
	Lordosis	0.0	0.2	0.0	0.0
	Vertebral fusion	0.0	3.4 *	4.2 *	4.8 **
	Vertebral anomaly	2.0	6.4 *	5.6	5.3
	Anomalous neural arch and/or spine	2.0	28.2 **	26.5 **	22.9 **
	Anomalous haemal arch and/or spine	3.3	5.2	4.1	6.3
**Haemal vertebrae**	Kyphosis	0.7	0.2	0.4	0.0
	Lordosis	0.0	0.0	0.0	0.0
	Vertebral fusion	3.3	0.8 **	2.7	0.5 **
	Vertebral anomaly	12.6	2.6 **	4.2 **	2.7 **
	Anomalous neural arch and/or spine	2.6	8.7 **	8.8 **	10.6 **
	Anomalous haemal arch and/or spine	7.9	4.9	9.7	10.4
**Caudal vertebrae**	Scoliosis	0.0	0.1	0.0	0.0
	Lordosis	2.0	0.0 **	0.0 **	0.2 *
	Vertebral fusion	0.0	0.5	0.3	0.2
	Vertebral anomaly	17.9	2.8 **	2.2 **	7.5 **
	Anomalous neural arch and/or spine	17.2	7.4 **	4.4 **	11.8
	Anomalous haemal arch and/or spine	9.9	1.2 **	1.0 **	4.3 *
**Caudal fin**	Anomalous hypural	1.3	0.4	0.2 *	0.2
	Anomalous epural	0.7	0.1	0.3	0.2
**Dorsal soft rays**	Anomalous pterygiophores	2.6	0.0 **	0.0 *	0.2 **
**Anal fin**	Anomalous pterygiophores	1.3	0.1 **	0.2 **	0.5
**Supernumerary bone**		0.7	0.0 **	0.1	0.0

Asterisks and double asterisks indicate significant (*p* < 0.05) and highly significant (*p* < 0.01) differences, respectively, compared with common carp.

**Table 3 vetsci-11-00030-t003:** Relative frequencies (%) of individuals affected by each type of anomaly.

Region	Type of Anomaly	Common Carp	Silver Carp	Grass Carp	Bighead Carp
**Cephalic vertebrae**	Kyphosis	0.0	3.4	6.8	0.0
	Lordosis	5.7	6.9	6.8	0.0
	Vertebral fusion	1.9	31.0 **	36.4 **	2.5
	Vertebral anomaly	1.9	20.7 **	25.0 **	2.5
	Anomalous neural arch and/or spine	11.3	54.0 **	93.2 **	55.0 **
	Anomalous haemal arch and/or spine	1.9	1.1	2.3	0.0
**Pre-haemal vertebrae**	Scoliosis	0.0	1.1	0.0	0.0
	Kyphosis	3.8	3.4	0.0	0.0
	Lordosis	0.0	2.3	0.0	0.0
	Vertebral fusion	0.0	23.0 **	50.0 **	15.0 **
	Vertebral anomaly	3.8	27.6 **	45.5 **	12.5
	Anomalous neural arch and/or spine	5.7	64.4 **	95.5 **	60.0 **
	Anomalous haemal arch and/or spine	7.5	16.1	27.3 **	25.0 *
**Haemal vertebrae**	Kyphosis	1.9	2.3	11.4	0.0
	Lordosis	0.0	0.0	0.0	0.0
	Vertebral fusion	9.4	5.7	27.3 *	5.0
	Vertebral anomaly	22.6	9.2 **	36.4	12.5
	Anomalous neural arch and/or spine	7.5	27.6 **	43.2 **	27.5 **
	Anomalous haemal arch and/or spine	13.2	14.9	59.1 **	30.0 *
**Caudal vertebrae**	Scoliosis	0.0	1.1	0.0	0.0
	Lordosis	5.7	0.0	0.0	2.5
	Vertebral fusion	0.0	4.6	6.8	2.5
	Vertebral anomaly	35.8	20.7 *	38.6	52.5
	Anomalous neural arch and/or spine	43.4	70.1 **	72.7 **	80.0 **
	Anomalous haemal arch and/or spine	26.4	11.5 *	15.9	35.0
**Caudal fin**	Anomalous hypural	3.8	4.6	4.5	2.5
	Anomalous epural	1.9	1.1	6.8	2.5
**Dorsal soft rays**	Anomalous pterygiophores	7.5	0.0	0.0	2.5
**Anal fin**	Anomalous pterygiophores	3.8	1.1	4.5	5.0
**Supernumerary bone**		1.9	0.0	2.3	0.0

Asterisks and double asterisks indicate significant (*p* < 0.05) and highly significant (*p* < 0.01) differences, respectively, compared with common carp.

**Table 4 vetsci-11-00030-t004:** Relative frequencies (%) of total anomalies observed in each body region.

Region	Common Carp	Silver Carp	Grass Carp	Bighead Carp
**Cephalic vertebrae region**	11.5	26.4 *	25.3 *	11.1
**Pre-haemal vertebrae region**	9.4	43.8 *	40.4 *	39.4 *
**Haemal vertebrae region**	24.5	17.2 *	25.8	24.2
**Caudal vertebrae region**	48.2	12.0 *	7.8 *	24.2 *
**Caudal fin**	2.2	0.5 *	0.4 *	0.5
**Other fins**	4.3	0.1 *	0.3 *	0.7 *

Asterisks indicate significant (*p* < 0.05) differences compared with common carp.

## Data Availability

The data presented in this study are available in the Excel file named ‘Appendix A’.

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
