# Peer review of "An Observational Study of Skeletal Malformations in Four Semi-Intensively Reared Carp Species"

_vetsci, 2024, doi:10.3390/vetsci11010030_

Round 1

Reviewer 1 Report

Comments and Suggestions for Authors

The topic is interesting, since there really are abnormalities that occur and cause harm to aquaculture. However, the possibility of malformations caused by possible inbreeding should be addressed as a possibility.

Reviewer 2 Report

Comments and Suggestions for Authors

The article describes the deformities present if four species of carp reared in semi intensive conditions, that apparently are not the ideal for some of the species. The study is just an assessment of deformities, but there are some concerns on the experimental design. There was a large variability inherent to the rearing of the studied species in optimal (common carp) and sub optimal conditions (the other 3 species) that can be the cause of deformities associated to different origins (natural vs induced spawning). There are no experimental replicates, since each species was reared in one pond. Can the environmental conditions of each pond affect the development and deformity incidence?

It would be advisable to provide the formulation of diets used and not just mention ingredients. 

Why were the herbivorous species fed with animal protein? This is causing a dietary unbalance that can induce the higher rates of deformities. Also common carp had a different diet as mentioned with cereal supplementation. 

The nomenclature must be revised (see notes in attached manuscript). The mention to fish age also must be clearly defined.

Specific comments can be found in the attached manuscript with reviewer notes

Comments on the Quality of English Language

English need minor checking

Reviewer 3 Report

Comments and Suggestions for Authors

The manuscript titled “Skeletal Malformations in Four Carp Species Reared in Semi-Intensive Systems” aims to whether the same rearing method can affect the development of skeletal deformities in four different carp species (common carp, silver carp, grass carp and bighead carp). The manuscript provides an anatomical description of the abnormalities which are present in these four different strains of carp. This study is very interesting, since it gives the opportunity to furtherly develop the rearing methods of the Albanian aquaculture industry. The findings are considered novel for these species and can be the first step to additional studies that can optimize the aquaculture industry of Albania. However, within the text it is mentioned in different sections of it, that the feeding protocol has to be optimized for the best rearing of these four species, no different rearing protocols/methods were examined. Thus, leaving a question whether this study was designed properly or sampled rearing populations for quality control to examine any skeletal deformities. Apart from that, the result section is poorly written with many repetitions of the same text, but with species-specific data. Similarly, high frequencies on severe abnormalities are observed, but the cases that are given in the Figure are mostly mild cases and cannot be considered representative cases. Also, the absence of statistic analysis is a critical point. Moreover, in the discussion section, the text in most of its length is a repetition of other studies results, without comparing them with this manuscript’s results. Finally, there are no conclusions about this study’s breakthrough and how it can assist the development of the Albanian aquaculture industry. Therefore, this manuscript cannot be recommended for publication in its present form and requires major revisions before that. Some major and minor comments follow.

Major Comments

1.     In the Material and Method Section there are some data missing. For example, what about the abiotic factors that take place during the rearing? Also, the size of the incubators and the rearing densities. A good suggestion is to include a paragraph section (maybe starting with that paragraph) and explain the rearing method in details and what were the parameters of the abiotic factors (e.g., temperature, salinity, oxygen saturation, pH, conductivity etc.) Here, it should be included also the diet protocol.

2.     L99-L103. Since the manuscript examines the effect of the diet on the development of skeletal deformities, it would be good to include the chemical composition of the diet (e.g. protein, dry matter, ash, phospholipids, total lipids). Then after these data are given, it would be good to examine whether such composition can be correlated to the different percentages of skeletal deformities among the different species.

3.     The Result section is very poorly written. The first paragraph makes reference to the Table 1 or S1, without describing the data. Usually, tables and figures are supportive elements to a text that has to be self-explained. Also, the paragraphs that describe the deformities in each species has repetitive text and the reader doesn’t understand any difference between them. Apart from that, general information about the observed deformities is given. In previous studies that you are referred (e.g., Bolgione et al., 2013b; Koumoundouros 2010; Prestinicola et al., 2013; Witten et al., 2005; etc.) there is a vast number of different types of skeletal deformities you could based on. Suggestion: Treat each species separately in the result section, where the text should be more descriptive. Based on the data, each specimen must have both different and similar deformity types between them. Examine and describe such deformities types based on previous studies. Is there any pattern concerning the types of deformities observed between the species that could be related to the rearing and the development of the species?

4.     The representative cases that were given in the Figure are mostly mild/light cases, whereas in the text they describe high frequencies of severe skeletal deformities, ranging between 3.0% and 27.9%, cases that should affect the external morphology. However, in the figure, only mild cases are presented and no differences could be identified based on the species. Suggestion: Have 1-2 cases of each severe deformity observed per species, with the normal specimen also as references.

5.     Statistic analyses are missing from this study. It would be good to have such analyses that could so if the observed differences are statistically significant (don’t forget to mention the use of statistics in the material and method section).

6.     L246-281: How this text is related to the results of this manuscript? It seems that data from other studies are only given, without being compared to the results of this study. These paragraphs should be re-written, having in mind the comparison between the results of this manuscript and how they are similar (or different) to previous studies. Also, these comparisons should lead to conclusions that can add additional value to the present study.

7.     L308-L314: How these sentences can be related to the results of the present study? Is there any correlation between the feeding habits, the diet stages and the development of the skeletal deformities? During rearing, the development of the different skeletal elements, take place in various developmental windows and not simultaneously? Is the development of skeletal deformities associated with the changes of the diets during rearing? This part is very interesting to be discussed, since it can show, how these results can assist in the optimization of the rearing methods for each of these four species.

8.     L318-L322: Like before, this statement can be furtherly discussed based on the results. A good point of discussion, but poorly written. If the feeding protocol is more suitable for suitable for common carp, then why the severe deformities in the bighead carp is similar to those of common carp (3.6 vs 3.0%).

9.     L345-351: The conclusions are not related to this study. Mainly, are assumptions based on previous studies that are not examined in the present study. When the factor of the spawning induction treatment was examined in relation to the development of skeletal abnormalities?

 Minor Comments

1. L51-L53: The economic impact of the development of skeletal deformities on European Aquaculture is estimated to be 50 million Euros per year as was stated by Hough (2009) in Bolgione et al. 2013b. Therefore, it would be good to include it here, and if such data for Albanian aquaculture exist, it would be good also to be stated in the introduction.

2. L63: Alcian-blue and Alizarin-Red

3. L120: replace the word “diaphanized” with the word “bleached”

4. L131-132: In what % of glycerol, were the specimen stored?

5. L239: replace the word “hatchery-produced” with the word “lab-reared”

6. L246-249: It is mentioned that the results are compared to other studies. However, no references are given for these “other studies”.

Comments on the Quality of English Language

No serious issues on the Quality of English Language were observed.

Round 2

Reviewer 3 Report

Comments and Suggestions for Authors

The manuscript in its present form is acceptable for publication. The manuscript is quite improved, especially the results section that was a repetition of the same text. No further revisions are required.

Author Response

We thank the reviewer for the positive evaluation of the revised manuscript.